# Development of Fenofibrate/Randomly Methylated β-Cyclodextrin-Loaded Eudragit^®^ RL 100 Nanoparticles for Ocular Delivery

**DOI:** 10.3390/molecules27154755

**Published:** 2022-07-25

**Authors:** Soe Yadanar Khin, Hay Man Saung Hnin Soe, Chaisak Chansriniyom, Natapol Pornputtapong, Rathapon Asasutjarit, Thorsteinn Loftsson, Phatsawee Jansook

**Affiliations:** 1Faculty of Pharmaceutical Sciences, Chulalongkorn University, 254 Phyathai Road, Pathumwan, Bangkok 10330, Thailand; soeyadanarkhin1993@gmail.com (S.Y.K.); haymansaunghninsoe@gmail.com (H.M.S.H.S.); chaisak.ch@chula.ac.th (C.C.); natapol.p@chula.ac.th (N.P.); 2Faculty of Pharmacy, Thammasat University, 99 Moo 18 Paholyothin Road, Klong Luang, Rangsit 12120, Thailand; rathapon@tu.ac.th; 3Faculty of Pharmaceutical Sciences, University of Iceland, Hofsvallagata 53, IS-107 Reykjavik, Iceland; thorstlo@hi.is

**Keywords:** cyclodextrin, fenofibrate, complexation, eye drops: nanocarriers, permeation

## Abstract

Fenofibrate (FE) has been shown to markedly reduce the progression of diabetic retinopathy and age-related macular degeneration in clinical trials and animal models. Owing to the limited aqueous solubility of FE, it may hamper ocular bioavailability and result in low efficiency to treat such diseases. To enhance the solubility of FE, water-soluble FE/cyclodextrin (CD) complex formation was determined by a phase-solubility technique. Randomly methylated-β-CD (RMβCD) exhibited the best solubility and the highest complexation efficiency (CE) for FE. Additionally, water-soluble polymers (i.e., hydroxypropyl methyl cellulose and polyvinyl alcohol [PVA]) enhanced the solubility of FE/RMβCD complexes. Solid- and solution-state characterizations were performed to elucidate and confirm the formation of inclusion FE/RMβCD complex. FE-loaded Eudragit^®^ nanoparticle (EuNP) dispersions and suspensions were developed. The physicochemical properties (i.e., pH, osmolality, viscosity, particle size, size distribution, and zeta potential) were within acceptable ranges. Moreover, in vitro mucoadhesion, in vitro release, and in vitro permeation studies revealed that the FE-loaded EuNP eye drop suspensions had excellent mucoadhesive properties and sustained FE release. The hemolytic activity, hen’s egg test on chorioallantoic membrane assay, and in vitro cytotoxicity test showed that the FE formulations had low hemolytic activity, were cytocompatible, and were moderately irritable to the eyes. In conclusion, PVA-stabilized FE/RMβCD-loaded EuNP eye drop suspensions were successfully developed, warranting further in vivo testing.

## 1. Introduction

Diabetic retinopathy (DR) is one of the most severe diseases affecting the microvasculature of the retina located at the back of the eye [1]. In 2020, the number of adults with DR was estimated to be 103.12 million, and by 2045, the number is expected to reach 160.50 million [2]. Age-related macular degeneration (AMD) is a serious eye disease that compromises the macular region of the retina and causes loss of central vision [3]. The global AMD prevalence is expected to reach 288 million people with the largest number of cases in Asia expected to reach approximately 113 million by 2040 [4]. DR and AMD are caused by several pathogenic factors and signaling pathways, such as vascular endothelial growth factor (VEGF), intercellular adhesion molecule-1 (ICAM-1), and tumor necrosis factor-alpha (TNF-α).

Fenofibrate (FE) is a fibric acid derivative that belongs to a class of lipid-lowering agents that are widely used to promote the lipid profiles of patients with metabolic disorders and dyslipidemia. In addition to its antilipidemic actions, FE acts as a peroxisome proliferator-activated receptor alpha (PPARα) agonist that has been reported to prevent the progression of DR and AMD in patients with diabetes. PPARα agonists inhibit the VEGF signaling pathway, which is involved in angiogenesis, inflammation, and the migration of endothelial cells [5]. Huang et al. (2021) investigated and found that topical administration of 0.1% and 0.5% FE nanoemulsion eye drops reduced the vascular leakage in DR and AMD models [6]. FE is categorized as Class II according to the Biopharmaceutical Classification System. The aqueous solubility of FE is <0.5 mg/L [7]. The limited aqueous solubility of FE may hamper its bioavailability in the treatment of such diseases. Among solubilization techniques, the formation of cyclodextrin (CD) inclusion complex is a promising technique to enhance the solubility of water-insoluble drugs.

CDs are cyclic oligosaccharides obtained by the enzymatic degradation of starch. The unique structure of CD includes a hydrophilic outer surface and a lipophilic inner cavity. Natural CDs include αCD, βCD, and γCD, which consist of six, seven and eight glucopyranose units, respectively. Owing to the limited aqueous solubility of natural CDs, CD derivatives, such as 2-hydroxypropyl-α-CD, sulfobutylether-β-CD (SBEβCD), randomly methylated-β-cyclodextrin (RMβCD), and 2-hydroxypropyl-γ-CD, have gained interest for various applications [8,9,10]. CDs can form inclusion complexes with lipophilic drugs by inserting their lipophilic moieties into the inner cavities of the CDs, leading to the increased solubility of aqueous solutions and improved bioavailability of drugs. In addition to solubility enhancement, complexation with CD can also promote drug permeation through biological membranes [11,12].

The addition of water-soluble polymers to drug/CD complexes improves the complexing and solubilizing efficiencies of CDs [13]. The ternary complex (i.e., drug/CD/polymer) can be obtained when the drug molecules are mixed with CD and polymer in solution, which provides the synergistic enhancement in drug solubility compared to without the polymer [14]. Recent studies reported that the complexation of hydrocortisone, dexamethasone, and naproxen with βCD can be increased by using polymers such as hydroxypropyl methylcellulose (HPMC) and polyvinylpyrrolidone [15]. Taupitz et al. (2013) investigated the ternary complexes consisting of CD derivatives (i.e., hydroxybutenyl-β-cyclodextrin and 2-hydroxypropyl-β-cyclodextrin [HPβCD]), and Soluplus^®^ (SOL) showed the highest solubilities of itraconazole [16]. The solubility of fexofenadine hydrochloride was increased by the ternary complexation with HPβCD and poloxamer [17]. A hydrophilic polymer, i.e., polyvinyl alcohol (PVA) was found to enhance the solubility and dissolution of diflunisal in a drug/CD complex [18].

Topical ocular drug delivery has been considered as an ideal route of administration for the treatment of ocular diseases. Most ocular diseases can be treated with conventional preparations, such as solutions, suspensions, and ointments. However, in ocular drug delivery, the major drawback of these conventional dosage forms is the failure to reach an optimal drug concentration at the site of action. This may be attributed to tear production, nasolacrimal drainage, transient residence time, corneal impermeability, and various anatomical and pathophysiological barriers predominant in the eye [19]. Nanoparticle (NP) platforms for ocular drug delivery have gained interest to overcome these obstacles.

Eudragit^®^ is a copolymer of ethyl acrylate, methyl methacrylate, and a low content ethacrylic acid ester with quaternary ammonium groups (trimethylammonioethyl methacrylate chloride) and has been widely used in ophthalmic preparations. Eudragit^®^ polymers are non-toxic and non-irritant. They have been classified according to their charges, such as cationic (Eudragit^®^ RL, Eudragit^®^ RS, Eudragit^®^ E), anionic (Eudragit^®^ L, Eudragit^®^ S), and neutral (Eudragit^®^ NE 30D, Eudragit^®^ 40D, Eudragit^®^ NM 30D). According to the literature, cationic Eudragit^®^ polymers, such as Eudragit^®^ RS 100 and RL 100, have a great potential in the ophthalmic drug delivery system because they exhibit no toxicity, high mucoadhesion, and controlled drug release [20]. Eudragit^®^ NPs have demonstrated the ability to encapsulate and enhance the solubility and bioavailability of poorly water-soluble drugs such as efavirenz [21], benznidazole [22], and quercetin [23]. It has been reported that drug-loaded Eudragit^®^ NPs did not show toxicity or irritation to ocular tissues [24].

Combining an inclusion drug/CD complex into various types of NPs has recently been used as a potential strategy for overcoming those drawbacks of each separate system. CDs-based NPs enable a unique drug delivery system that combines the benefits of both components, i.e., CDs improve the aqueous solubility of drugs and drug loading, while NPs provide the targeted drug delivery [25]. The aim of this study was to investigate FE solubility through CD inclusion complexes and developed FE-loaded Eudragit^®^ NP (EuNP) eye drop formulations. The combined strategies (i.e., FE/CD inclusion complexes and FE-loaded EuNP) were optimized and used to develop the formulations. The physicochemical and chemical properties of FE/CD-loaded EuNP formulations were determined. In vitro mucoadhesion and in vitro permeation through artificial membranes were performed. In addition, in vitro toxicity and cell viability (CV) were evaluated.

## 2. Results and Discussion

### 2.1. Thermal Stability

Table 1 shows the percentage of FE remaining in aqueous 2.5% *w*/*v* or 5% *w*/*v* HPβCD solution after zero to three heating cycles. HPβCD has been reported to increase FE solubility; therefore, it was selected for this study [26]. The thermal stability of FE in pure water (in the absence of HPβCD) was not evaluated because of its lower limit of detection. After autoclave heating (121 °C for 20 min), the percentages of FE remaining in aqueous 2.5% (*w*/*v*) and 5% (*w*/*v*) HPβCD solutions were approximately 90% and 92% after three cycles, respectively. FE degradation may be related to the hydrolysis of the ester group in FE. In addition, FE was remarkably degraded upon exposure to relatively high temperatures [27]. Conversely, FE was relatively stable after three heating cycles in an ultrasonic bath (at 60 °C for 30 min). Since drug degradation during sonication was <2% after three cycles, the heating method was selected to determine the solubility of FE in aqueous CD solutions.

### 2.2. Solubility Determination

#### 2.2.1. CD Solubilization of FE

Figure 1 shows the phase-solubility diagrams of FE in the aqueous solutions of parent CDs (αCD, βCD, and γCD) and βCD derivatives (HPβCD, SBEβCD, and RMβCD). According to the Higuchi and Connors classification, αCD exhibits an A_L_-type phase-solubility profile, which demonstrates that the solubility of FE linearly increases with αCD concentrations, forming a 1:1 complex (Figure 1a). In βCD and γCD, FE solubility increased with increasing CD concentrations, starting at low CD concentrations then leveling off to higher CD concentrations. This may be due to the limited solubility of FE/CD complexes, which represent B_S_-type phase-solubility profiles (Figure 1b,c).

Table 2 shows the apparent complexation constants (K_1:1_, K_1:2_) and the complexation efficacy (CE) values of the FE/CD complexes. The K_1:1_ and CE values of βCD and γCD were calculated from the linear regions of the phase-solubility profiles. βCD solubilization in FE was almost 10 times higher than that of αCD and γCD. Based on the characteristics and cavity sizes of CDs, αCD can form complexes with lower molecular weights or aliphatic side-chain compounds. Meanwhile, βCD can accommodate aromatic and heterocyclic molecules, and γCD has an affinity for larger molecules, such as macrocycles and steroids [28]. However, parent βCDs have limited solubility in aqueous media. Thus, βCD derivatives (i.e., SBEβCD, HPβCD, and RMβCD) were selected for further solubility determination of FE.

SBEβCD displayed an A_L_-type phase-solubility profile, whereas HPβCD and RMβCD displayed A_P_-type profiles (Figure 1d). FE/RMβCD had the highest CE value among the tested βCD derivatives (Table 2). To the best of our knowledge, the complex formation of FE and RMβCD has not been studied. When βCD hydroxyls are methylated, hydrogen bonding is disrupted. This results in macrocyclic ring deformation and decreased preference for water interaction, allowing a favorable interaction with guest molecules [29]. The addition of methyl groups to RMβCD may improve its solubility by extending the hydrophobic portion of the CD cavity, enabling and stabilizing FE inclusion complexes. [30,31]. Thus, based on the phase-solubility study results, RMβCD was selected for further study.

#### 2.2.2. Effect of Water-Soluble Polymers on RMβCD Solubilization in FE

Water-soluble polymers can enhance the CD complexation of drugs and drug permeation through biological membranes, possibly through the formation of ternary complexes or co-complexes. [32,33,34,35]. In this study, we investigated the effect of nonionic water-soluble polymers (i.e., SOL, poloxamer 407 [P407], HPMC, and PVA) on RMβCD solubilization in FE. Figure 2 shows the different types of hydrophilic polymers affecting the solubility of FE/RMβCD complexes. The combination of a polymer and RMβCD exhibited an additive effect in enhancing the solubility of FE compared to the binary complex (i.e., FE/RMβCD complex). The highest FE solubility was observed in HPMC and PVA, followed by SOL and P407. HPMC enhances the solubility of drug/CD complexes by forming ternary complexes [36,37,38]. This may be attributed to the strong interaction between the hydrophobic parts of HPMC and the crystal habit of FE during aggregate formation. Thus, the crystalline drug was altered to an amorphous form, resulting in increased drug solubility [39]. PVA has abundant hydrogen bond donors, while FE has four hydrogen bond acceptors. PVA has a strong potential to form hydrogen bonds with FE molecules, allowing for an additive effect on RMβCD solubilization in the drug [40]. In P407 and SOL, the solubility of FE was insignificantly enhanced by increasing polymer concentrations. The hydrophobic parts of both amphiphilic copolymers were possibly inserted into the CD cavity, competing with drug molecules for CD affinity and resulting in a negligible CD solubilizing effect [41]. HPMC and PVA were chosen to develop FE eye drop preparations based on their ability to increase the solubility of FE/RMβCD complexes.

### 2.3. Characterization of FE/RMβCD Complexes

#### 2.3.1. Solid-State Characterization

Fourier transform infrared (FT-IR) spectroscopy was applied to verify the interactions between CD and guest molecules by observing the changes or shifts in the absorption spectrum [42]. The FT-IR spectra of intact FE, RMβCD, and the physical mixture (PM) and the freeze-dried (FD) of its binary complex, are displayed in Appendix A. The characteristic peaks of pure FE were observed at 1726.52 cm^−1^ and 1649.62 cm^−1^, which corresponded to two C=O frequencies of the ester and ketone group, respectively (Appendix A). In addition, isopropyl -CH(CH_3_)_2_ presented as a small peak at 1384.71 cm^−1^, symmetrical strain C-C=C showed aromatic rings at 1598.03, 1501.52 cm^−1^, and C-Cl stretching was observed at 764.59 cm^−1^. These IR characteristic peaks were in accordance with those previously reported [43,44,45]. The FT-IR spectrum of RMβCD showed a broad absorption band at 3388.52 cm^−1^ due to -OH stretching (from residual non-substituted groups), and a large number of bands and a distinct peak were observed below 2000 cm^−1^ related to coupled modes from the CD rings (Appendix A) [46].

In the FT-IR spectrum of the PM sample (Appendix A), the peaks of FE at 1726.52, 1649.62, 1384.71, and 1598.03 cm^−1^ were somewhat shifted to 1727.52, 1649.70, 1362.63, and 1598.45 cm^−1^, respectively. In addition, the peak of RMβCD at 3388.52 was slightly shifted to 3387.14 cm^−1^. This indicated that there was less interaction between FE and RMβCD with a simple superimposition of individual components. In case of the FT-IR spectrum of the FD sample (Appendix A), all distinct peaks of FE had disappeared, and no new peaks were formed. This may be possible due to some interactions between functional groups of FE and functional groups in the hydrophobic cavity of RMβCD during the formation of an inclusion complex.

Powder X-ray diffraction (PXRD) is one of the most important characterization tools used to detect the CD complexation of a compound in solid states [42]. The PXRD patterns of intact FE and RMβCD, and their PM and FD samples are shown in Appendix A. The diffractogram of FE displays the sharp intense peaks at 14.40, 16.20, 22.22, 26.22, 29.05, and 30.34°, which represented the crystallinity of FE (Appendix A) [43,47].

The diffractogram of RMβCD did not show any sharp peak and only a halo pattern was observed, which indicated the nature of the amorphous form (Appendix A) [48]. Most of the distinct peaks of FE were detectable in the diffraction patterns of PM of the FE/RMβCD complexes (Appendix A). This suggested that there was no interaction between the pure FE and RMβCD. On the contrary, the crystal diffraction peaks of FE had completely disappeared and showed the broad pattern in the FD sample (Appendix A). This indicated that the crystalline nature of FE was transformed to an amorphous state, which was probably due to the formation of a drug/CD complexation. Similarly, Ding et al. (2018) reported the disappearance of the distinct peaks of FE via inclusion complex formation of FE/HPβCD [43].

The differential scanning calorimetry (DSC) thermograms of FE and RMβCD, and their PM and FD samples are shown in Appendix A. The DSC thermogram of pure FE (Appendix A) showed a sharp endothermic peak at 82.47 °C, which is indicative of its melting point. This observation was in accordance with the previous reports [26,43]. The thermogram of RMβCD (Appendix A) revealed no sharp endotherm, confirming its amorphous nature. However, a broad descending curve was observed at 71.26 °C, which attributed to the loss of water molecules in the CD cavity. Due to dehydration in the CD cavity, a DSC thermogram of amorphous CD showed broad endothermic peaks at about 90–130 °C [42].

The sharp endothermic peak of the FE in PM FE/RMβCD was slightly shifted to a lower temperature and appeared at 80.28 °C (Appendix A). These findings could be explained by the presence of weak or no interaction between the pure components in the PM, or by the possibility of an interaction between FE and RMβCD induced by the DSC heating process. In case of FD FE/RMβCD (Appendix A), the DSC thermogram showed the disappearance of the endothermic peak characteristic of the FE, thus suggesting inclusion complex formation between FE and RMβCD in the solid state.

#### 2.3.2. Solution-State Characterization

Proton nuclear magnetic resonance (^1^H-NMR) spectroscopy was used to investigate the interactions between FE and RMβCD in DMSO-*d_6_*. Table 3 shows the changes in the chemical shifts (Δδ*) of FE alone and in the presence of RMβCD. The ^1^H signal of the inner protons, i.e., the H3′ and H5′ protons of CD, is essential for the possible complex formation between the drug and CD [49]. In the RMβCD cavity, the upfield shift of H3′ was −0.001 ppm, and the downfield shift of H5′ was +0.002 ppm, showing that the Δδ* value of H5′ was higher than that of H3′. Therefore, FE was deeply inserted into the hydrophobic cavity of RMβCD. A downfield shift of −0.002 was also observed in H6′ and CH_3_OC6, implying that the formation of FE/CD interactions can also occur in the narrow rim of the RMβCD cavity.

For FE protons, the Δδ* of H1, 2 (CH_3_-) and H3 (-CH-O) in the isopropyl connected to the ester group showed an upfield shift of −0.002 ppm. Additionally, the Δδ* of H12,13 (aromatic ring protons attached to the Cl group) showed a downfield shift of +0.003 ppm, indicating that hydrogen bonding interactions with RMβCD primarily occurred in the protons of the methyl groups, followed by that of the aromatic rings. This result is similar to that of the studies of Ding et al. (2018) and Do Thi (2011); they found that the isopropyl group and aromatic rings of FE are inserted into the hydrophobic inner cavity of HPβCD [43,50].

Figure 3 shows the expanded 2D Rotating-Frame Overhauser Enhancement Spectroscopy (ROESY) spectrum of the FE/RMβCD complex. A correlation was found between the H1, H2, and H3 (i.e., the isopropyl connected to the ester group) protons of FE and the H5′ proton of RMβCD, proving that the complex is located near the narrow rim of RMβCD. The H8, H9, H10, and H11 protons of FE had a significant cross peak overlap with the H3′ and H5′ protons of RMβCD. Meanwhile, the H12 and H13 protons of FE only had a cross peak with the H5′ proton of RMβCD. These cross peaks show that the H12 and H13 protons of FE were completely inserted into the cavity of RMβCD. Additionally, the inner aromatic rings (i.e., H6, H7, H8, and H9) of FE had other cross peaks with CH_3_OC6 at the methoxy group of RMβCD. These data confirmed that the aromatic ring attached to Cl was located within the RMβCD cavity, while another ring (i.e., an inner aromatic ring) interacted with the additional binding site of the methoxy group.

#### 2.3.3. Molecular Docking and Molecular Dynamic Simulation

Molecular docking was conducted to investigate the preferential binding mode of the FE/RMβCD complex. Figure 4 displays the energy-optimized modes of the FE/RMβCD complex, using AutoDock. The FE molecule exhibited two possibly favorable orientations inside the RMβCD cavity. The aromatic ring attached to the Cl atom, inserted into the RMβCD cavity, was labelled as form I (Figure 4a). Meanwhile, the isopropyl group of the FE insertion was labelled as form II (Figure 4b). The docking interaction energies of forms I and II are −7.2 and −6.0 kcal/mol, respectively. According to the results, form I was predominantly inserted into the RMβCD cavity, suggesting that the aromatic ring attached to the Cl atom was the preferrable binding mode for FE, rather than the isopropyl group. The FE/RMβCD complexes were further analyzed using molecular dynamics simulations. The simulation resulted in 500 trajectories for 100 ps, demonstrating the movement of FE in the RMβCD cavity. The first 100 trajectories were removed as burn-ins based on the total energy plot. Trajectories 300 to 500 were overlayed using 20 frames for smoothing, as shown in Figure 4c,d. The results showed that the simulations had a good correlation with the experimental ^1^H-NMR and 2D ROESY data.

Figure 5 shows the proposed conformational structure of FE/RMβCD inclusion complexes. The stoichiometry of the FE/RMβCD complexes were 1:1 and/or 1:2. FE has a relatively long molecular structure composed of two aromatic rings, a hydrophobic organic ether, and an ester chain. Thus, one FE molecule may form an inclusion complex with two RMβCD molecules. It can be inferred that each CD occupied a different end of the FE molecule (i.e., half of the FE was inserted into the inner cavity of one monomer of RMβCD and the other half was embedded in another monomer of RMβCD). The structure was held in place by the formation of hydrogen bonds between the hydroxyl or methoxy groups of RMβCD and the chlorine atoms and isopropyl groups of FE. This corresponded with FE/RMβC displaying A_P_-type diagrams.

### 2.4. Physicochemical and Chemical Characterization of FE-EuNP Eye Drop Formulations

Prior to loading the FE into the NPs, drug-free EuNPs were screened. EuNPs consisting of Eudragit^®^ RL 100 (200 and 400 mg) with 1% (*w*/*v*) PVA or 0.1% (*w*/*v*) HPMC had an appropriate particle size (100–200 nm)**,** narrow polydispersity index (PDI), and high zeta potential value. Table 4 shows the physicochemical and chemical properties of FE eye drop formulations. The pH values of all formulations were 7.41–7.48, which is in an acceptable range for the ideal pH of eye drops [51]. The tonicity of the ophthalmic preparations was close to that of the lacrimal fluid (260–330 mOsm/kg). All formulations had a low viscosity ranging from 2 to 4 mPa·s.

The particle sizes of all Eudragit^®^-based formulations were in nanometers. The entrapped drug in the form of the FE/RMβCD inclusion complexes slightly decreases the particle size of the EuNPs compared to that of the non-complex ones. Similar results have been reported by Trapani et al. (2010) [52]. This may be attributed to the surface-active properties of methylated βCD [53,54] that can reduce the NP curvature. Water-soluble polymers also influence particle size. The particle sizes of the formulations containing PVA were smaller than those containing HPMC. The rigid structure of long chain of HPMC makes looping around the NPs more difficult than with the single chain of PVA [55]. The PDI values of all formulations (0.20–0.28) were <0.3, which is within the acceptable range, indicating uniform particle distribution [56].

All FE-loaded EuNP eye drop formulations exhibited positive zeta potential values ranging from +22.23 mV to +41.92 mV. These values were obtained from the characteristics of Eudragit^®^, suggesting that Eudragit^®^ RL 100 was mainly located on the surface of the particles [57]. The zeta potential values of F5 and F7 (containing 4% Eudragit^®^ RL 100) were significantly higher than that of the other FE eye drop formulations (*p* < 0.05). Positive surface charge is a favorable property of NPs intended for ophthalmic drug delivery. It can facilitate adequate adhesion to the corneal surface and it strongly interacts with the negatively charged mucosa of the conjunctiva and anionic mucin, prolonging the residence time of the formulations [20].

The percentage of encapsulation efficiency (%EE) of the formulations containing RMβCD significantly increased by 1.5 times compared to those without RMβCD (*p* < 0.05), indicating that incorporating RMβCD into polymeric EuNPs could increase %EE and drug loading. Lopedota et al. (2009) [58] found that a CD inclusion complex with glutathione improved the encapsulation capacity of EuNPs. Further, using PVA as a stabilizer (F1 and F3) resulted in a slightly higher %EE than HPMC (F2 and F4). This may be because PVA provided a greater additive effect on RMβCD solubilization in FE than that of HPMC. Ternary FE/RMβCD/PVA complexes further enhanced the %EE of FE in the EuNPs. In our approach, >83%EE was expectedly achieved in F5, which contained a high Eudragit^®^ RL 100 concentration, in the presence of RMβCD and PVA. In suspensions, the %EEs of F6 and F7 were slightly higher than that of their corresponding FE dispersions (i.e., F3 and F5). Increasing %EE was possible because inducing applied energy from a high pressure homogenizer (HPH) generates more encapsulated particles.

In the FE-loaded EuNP eye drop suspensions, the particle size of insoluble solid particles was determined. Solid particles in acceptable ophthalmic suspensions should be <10 µm in diameter to decrease irritation or prevent eye scratching, thereby providing patient comfort [59,60]. In this study, the HPH technique was applied to reduce the size of solid particles in suspension. The solid particle sizes of F6 and F7 were 8.20 ± 0.77 µm and 8.78 ± 0.20 µm, respectively, which were acceptable for eye drop suspensions [60].

### 2.5. In Vitro Mucoadhesive Studies

To evaluate the interactions between mucin and FE-loaded EuNP eye drop formulations, the percentage (%) of mucoadhesion in all formulations was determined by UV spectrophotometry, as shown in Figure 6. The results revealed that formulations containing higher concentrations of Eudragit^®^ RL 100 had a higher %mucoadhesion. This may be due to the establishment of electrostatic interactions between the ammonium groups of Eudragit^®^ and the negatively charged domains of the mucin chains of sialic acid and sulfated sugar molecules. PVA-based FE eye drop dispersions (F1 and F3) had significantly better mucoadhesive properties than that of the corresponding HPMC-based formulations (F2 and F4) (*p* < 0.05). PVA can form strong hydrogen bonding interactions with mucins because it possesses a larger number of hydroxyl groups than HPMC. The relatively inflexible main chain of HPMC may also make wrapping around the NP droplets more difficult than in the single chain of PVA [55].

The %mucoadhesion of FE eye drop formulations containing RMβCD was slightly higher than that of formulations without RMβCD. This may be due to the additional hydrogen bond formation between the hydrophilic outer part of the CD molecules and the -OH of sugars and other O- and N-containing groups of mucin molecules. Therefore, the synergistic action of CD and Eudragit^®^ can enhance the mucoadhesive properties [58]. In particular, FE formulations with high FE and Eudragit^®^ RL 100 content, together with the presence of RMβCD stabilized by PVA (F7), provided the highest binding capacity with mucin. Based on these results, PVA-stabilized FE/RMβCD-loaded EuNP eye drop suspensions (F6 and F7) were selected for further study.

### 2.6. In Vitro Release Studies

The cumulative FE release profiles of selected FE eye drop formulations are shown in Appendix A. The ranking order of the release rate of FE at the initial phase (i.e., 0–6 h) is F7 > F5 > F6 > F3. It demonstrated that the amount of drug in the formulation directly impacted the drug release profiles. According to Fick’s law, the greater the dissolved drug, the higher drug release rates were obtained. In addition, it was found that the release rate was related to the Eudragit^®^ RL100 content. Both the FE dispersions and suspensions containing 4% (*w*/*v*) Eudragit^®^ RL100 provided a significantly higher drug release than those of the corresponding formulations comprised of 2% (*w*/*v*) Eudragit^®^ RL100 (*p* < 0.05). It is possible, due to the characteristic of Eudragit^®^ RL100, that a copolymer of acrylic and methacrylic acid esters with hydrophilic ammonium groups, that promote water permeability by acting as a channeling agent for entry of liquid medium through the dispersed NPs wall, led to polymer swelling and consequently to drug release [61]. This finding is consistent with the results of other investigations [62].

The drug release was affected by a combination of polymer swelling, erosion, and diffusion through the hydrated matrix. The drug release data for FE/RMβCD-loaded Eudragit^®^ NPs were fitted to four kinetic models, such as zero order, first order, Higuchi, and Korsmeyer–Peppas, to determine the release constant and regression coefficients (R^2^) (Appendix A). Among the models tested, the kinetic results showed that the drug release profiles for all formulations were best fitted with the Korsmeyer–Peppas model based on the regression coefficients (R^2^ ≥ 0.96). The release exponent “*n*” value of F3 was 0.86, and it was within the range of 0.45–0.89. This indicated non-Fickian diffusion or anomalous diffusion, and it considered a drug diffusion mechanism or diffusion coupled with erosion [63]. Thus, according to the drug release rate result of F3, the drugs might be slowly released with a diffusion mechanism from NPs and, hence, prolong the release of drugs among the tested formulations. On the other hand, the n values of F5, F6, and F7 were above 0.89. This indicated super case II transport and can be considered as erosion of the polymer chain. It is supported that the solvent penetration velocity is high, which could enhance the transportation of the medium to the polymeric matrix [64]. Therefore, the drug release rates of these three FE formulations were higher than that of F3. According to the high FE release rate followed by the slow release and great mucoadhesive characteristics obtained from F6 and F7, these two FE eye drop formulations were chosen to study in vitro permeation through artificial membranes.

### 2.7. In Vitro Permeation Study

The dual membrane, which consists of a hydrophilic cellophane membrane on the receptor side and a fused octanol membrane on the other, is easy to use and has similar characteristics to that of artificial membranes. It can also be modified by adding a mucin layer on the donor side (mucin-coated octanol membrane) to simulate the biological membrane. Table 5 displays the permeation flux (*J*) and the apparent permeation coefficient (*P*_app_) of FE-loaded EuNP suspensions through octanol dual and mucin-coated octanol membranes. The rate of drug diffusion through a membrane is highly controlled by the drug concentration gradient between the exterior and interior membrane. The *J* of F7 was significantly higher than that of F6 (approximately twofold) because F7 has a higher content of dissolved FE. However, no significant difference was found between their permeation rates. Compared to that of an uncoated octanol membrane, FE permeation through mucin-coated octanol membrane yields slightly lower *J* and *P*_app_ values for F6 (2% Eudragit^®^). In contrast to that of F7 (4% Eudragit^®^), both permeability parameters were significantly lower (*p* < 0.05) in F6. EuNPs are recognized for their mucoadhesive properties, such as the electrostatic interaction between the positively charged nanosized formulation and negatively charged mucin-coated membrane. The higher the amount of Eudragit^®^, the higher the %mucoadhesion with mucin, thereby decreasing the *J* and *P*_app_ of F7 through the mucin-coated octanol dual membrane.

### 2.8. Morphology

Figure 7 shows the transmission electron microscopic (TEM) and scanning electron microscopic (SEM) images of FE-loaded EuNP suspensions (F6 and F7). The TEM images of EuNPs containing FE showed that the particles were uniformly distributed and presented a spherical shape with irregular surfaces. This demonstrated that FE encapsulation in 2% Eudragit^®^ (F6) has a smoother surface and slightly smaller particle size than in 4% Eudragit^®^ (F7) (Figure 7a and Figure 7b, respectively). Additionally, the NPs of F7 were closely packed and tended to form agglomerates. The particle sizes of the EuNP-based suspensions corresponded with the mean diameters estimated by the dynamic light scattering (DLS).

The scanning electron microscopic (SEM) images of FE solid particles in FE-loaded EuNP suspensions (F6 and F7) revealed rectangular FE crystals. The FE solid particle sizes in F6 were approximately 8 µm (Figure 7c) in diameter, which correlated with the solid particle sizes obtained by optical light microscopy. In contrast, most of the FE solid particles in F7 were approximately 9 µm in diameter, and some particles were >10 µm (Figure 7d). A particle size > 10 µm in diameter may lead to a foreign body sensation in the eye following ocular application. SEM indicated the presence of larger particles, which were not observed under light microscopy. This can be attributed to subsampling and particle agglomeration during the SEM sample preparation.

### 2.9. In Vitro Hemolytic Activity

Figure 8 displays the %hemolysis of FE/RMβCD-loaded EuNP suspensions and their corresponding blank formulations, including aqueous solutions of FE/RMβCD complexes. The FE/RMβCD complexes displayed approximately 30% hemolysis at a concentration of 1000 μg/mL. This may be because FE itself and RMβCD, a lipophilic CD, can cause hemolysis, even at low concentrations [65]. Hemolysis of FE/RMβCD-loaded EuNP suspensions (F6 and F7) was observed at a percentage of 15–18%, even at an FE concentration of 1000 μg/mL, which was twofold lower than that of the FE/RMβCD complexes. The shielding effect of EuNPs resulting from the association of FE/RMβCD with Eudragit^®^ could be attributed to reduced toxicity. Moreover, EuNPs regulated drug release, leading to a less likely binding with the RBC membrane and consequently, lowered toxicity. In fact, no hemolysis was observed in the two drug-free EuNPs across the same concentration range (10–1000 μg/mL). This indicated that the excipients used in the FE eye drop formulations were not cytotoxic [66,67]. As the FE concentration increased, hemolytic activity increased in a dose-dependent manner.

### 2.10. Irritation Study by Hen’s Egg Test on Chorioallantoic Membrane (HET-CAM) Assay

Using the HET-CAM test, the potential ocular irritancy of FE/RMβCD-loaded EuNP suspensions (i.e., F6 and F7) was evaluated with their corresponding drug-free formulations (blank). This test is an alternative to the Draize test, which is used to evaluate ocular formulations [68,69]. The irritation scores (IS) of the samples mentioned above were compared with those of C+ and C−. The test is valid if the negative control does not induce irritation and the positive control causes severe irritation. The IS of C+ was 17.0 ± 0.0, indicating strong irritation (Figure 9a), while the IS of C− was 0 (Figure 9b). None of the blank formulations induced hemorrhage, lysis, or coagulation, and an IS of 0.0 (Figure 9c,d) was recorded. Applying FE/RMβCD-loaded EuNP suspensions (F6 and F7) did not result in any visible sign of irritation or vascular damage, indicating that the developed formulations were safe for ocular drug delivery (Figure 9e,f).

Overall, the higher the Eudragit^®^ concentration in the FE eye drop suspensions (F7), the more evidence of particle aggregation was found. In addition, FE solid particles were larger in size and a higher %hemolysis was observed at high concentrations of FE (up to 800 µg/mL). Therefore, only F6 was considered for a further in vitro cytotoxicity assay.

### 2.11. Cell Viability (CV) and Short-Time Exposure (STE) Test

We determined the in vitro cytotoxic effect of the selected FE/RMβCD-loaded EuNP suspension (i.e., F6) on the Statens Serum Institut Rabbit Cornea (SIRC) cell lines. CV < 70% in the methylthiazolyl-diphenyl-tetrazolium bromide (MTT) assay was declared cytotoxic according to the UNI ISO 10993-5 standard [70,71]. The FE eye drop suspension was incubated with the cells at concentrations ranging from 1 to 400 µg/mL for 24 h, and no cytotoxicity (>70% CV) was observed (Figure 10). Although the lipophilic nature of methylated CD can interact with the cell membrane and induce cell death, the cells were shielded by the Eudragit^®^ network from this effect by forming CD-based polymeric NPs, resulting in lower cytotoxicity to the cells. This result suggested that eye drops containing Eudragit^®^ RL 100 are biocompatible and may potentially regulate drug delivery in biomedical applications, particularly ocular administration.

In addition to the cytotoxicity study, we investigated in vitro eye irritation using the STE test, which can provide information similar to that of the Draize test on rabbits [71]. The results revealed that SIRC cells can survive treatment with FE/RMβCD-loaded EuNPs. Table 6 shows the %CV of the SIRC cells after 5 min of exposure to 5% and 0.05% F6. The scores represent eye discomfort. The overall score for ocular irritation potential was 2. Based on these results, FE/RMβCD-loaded EuNPs were classified as moderate irritants, suitable for ocular drug delivery.

## 3. Materials and Methods

### 3.1. Materials

The FE was donated by Siam Bheasach Co., Ltd. (Bangkok, Thailand). The αCD, βCD, and γCD were given by Ashland (Wilmington, DE, USA). HPβCD with 0.65 molecular substitution (MS) (molecular weight [MW] = 1400 Da) and SBEβCD with 0.9 MS (MW = 2163) were donated by Roquette (Lestrem, France). RMβCD with 1.8 MS (MW = 1312 Da) was purchased from Wacker Chemie AG (Burghausen, Germany). Eudragit^®^ RL 100 was provided by Evonik Industries AG (Marl, Germany). P407 and SOL (a polyvinyl caprolactam-polyvinyl acetate-polyethylene glycol graft copolymer) were obtained from BASF SE (Ludwigshafen, Germany). Benzalkonium chloride (BAC), ethylenediaminetetraacetic acid disodium salt (EDTA), sodium chloride, HPMC (MW = 4000 Da), PVA (MW = 27,000 Da), cellulose acetate (CA), *n*-dodecanol, and mucin from porcine stomach type II were purchased from Sigma-Aldrich (St. Louis, MO, USA). A semi-permeable membrane (molecular weight cut off [MWCO] = 12–14 kD) was purchased from Spectrum Labs (Breda, The Netherlands). Commercial sheep blood was purchased from Chulalongkorn University Laboratory Animal Center (Bangkok, Thailand). The SIRC cell line was purchased from the American Type Culture Collection (ATCC) (Manassas, VA, USA). All cell culture reagents were purchased from Invitrogen (Thermo Fisher Scientific; Waltham, MA, USA). All other chemicals used were of analytical reagent grade purity. Milli-Q water (Millipore, Billerica, MA, USA) was used to prepare all solutions.

### 3.2. Thermal Stability Study

A heating method was applied to accelerate drug/CD inclusion complex formation [37]. The thermal stability of FE in aqueous CD solutions was studied by two heating methods: autoclaving and sonication. A small amount of FE was dissolved in an aqueous HPβCD solution (2.5% or 5% *w*/*v*) that was divided into four sealed vials. To achieve equilibrium, the prepared solution was constantly agitated in a shaking incubator at 30 ± 1 °C for 24 h. Then, the samples were heated in an autoclave at 121 °C for 20 min during zero, one, two, and three heating cycles. The analogue set was placed in a sonicator at 60 °C for 30 min. FE content was determined by high-performance liquid chromatography (HPLC). Each sample was analyzed in triplicate.

### 3.3. Solubility Determination

An excess amount of FE was added to aqueous solutions containing αCD (0–12% *w*/*v*), βCD (0–1.5% *w*/*v*), and βCD derivatives, namely, HPβCD, SBEβCD, RMβCD (0–10% *w*/*v*), or γCD (0–15% *w*/*v*). The drug suspensions, in sealed vials, were heated in a sonicator at 60 °C for 30 min and allowed to cool at room temperature. The suspensions were equilibrated for seven days at 30 ± 1 °C under constant agitation. After reaching equilibrium, the suspensions were filtered through a 0.45-µm nylon filter and analyzed by HPLC [37]. Phase-solubility diagrams were constructed by plotting the total dissolved FE concentration (M) against CD concentration (M). The apparent complexation constant (K_1:1_ and/or K_1:2_) of the FE/CD complex was determined according to the phase-solubility method [72] and the CE was calculated by following Equations (1)–(3) [73].
(1)$$K_{1:1}=\frac{\mathrm{slope}}{S_{0}\left(1-\mathrm{slope}\right)}$$
(2)$$\left[S_{t}\right]-\left[S_{0}\right]=K_{1:1}\left[S_{0}\right]\left[\mathrm{CD}\right]+K_{1:1}\cdot K_{1:2}\;\cdot \;S_{0}\left[\mathrm{CD}\right]2^{\;}$$
(3)$$\mathrm{CE}=\frac{\mathrm{slope}}{\left(1-\mathrm{slope}\right)}=K_{1:1\;}\cdot \;S_{0}$$
where S_0_ is the intrinsic solubility of FE and S_t_ is the total amount of dissolved FE.

The effect of water-soluble polymers on RMβCD solubilization by FE was further investigated. The solubility of FE in aqueous 5% (*w*/*v*) RMβCD solutions was briefly determined in the presence of water-soluble polymers (i.e., SOL, P407, HPMC, and PVA). Polymer concentrations were used in the range of 0.01–1% *w*/*v*. The excess amount of FE was added to the complexing medium and then heated in a sonicator at 60 °C for 30 min. The suspensions were equilibrated for seven days at 30 ± 1 °C under constant agitation. After attaining equilibrium, the suspensions were filtered through a 0.45 µm nylon filter and analyzed by HPLC.

### 3.4. Quantitative Analysis of FE

The quantitative determination of FE was performed on Agilent 1260 Infinity II (Agilent Technologies, Inc., Santa Clara, CA, USA), a reversed-phase HPLC component system, consisting of a liquid chromatography pump (quaternary pump, G7111B), a column compartment (G7116A), an ultraviolet–visible spectroscopy detector (G7115A), and an auto sampler (G7129A) with Chem Station Software Version E.02.02. The HPLC settings were as follows:
Mobile phaseAcetonitrile and 0.05% phosphoric acid (75:25 [*v*/*v*])Chromatographic columnShiseido^TM^ Capcell Pack C18 MG II S-5, 150 × 4.5 mm ID with C18 guard cartridge column MGII 5 µm, 4 × 10 mmFlow rate1.0 mL/minOven temperature30 °CUV detector wavelength286 nmInjection volume10 µL

### 3.5. Preparation and Characterization of the FE/RMβCD Complex

#### 3.5.1. Solid-State Characterization

Aqueous solution containing a 1:1 molar ratio of the binary complexes, i.e., FE/RMβCD, were prepared by heating in a sonicator at 60 °C for 30 min. The sample was equilibrated at 30 ± 1 °C for 24 h under constant agitation. After 24 h, the sample was centrifuged (Thermo Fisher Scientific, Waltham, MA, USA) at 13,000 rpm for 20 min. Then, the supernatant was withdrawn, frozen at −80 °C for 2 h, and lyophilized at −52 °C for 48 h in a freeze dryer (Labconco Corporation, Kansas City, MO, USA) to obtain a solid complex FD. Identical PM was prepared by careful blending of ingredients in a mortar with pestle. The samples were characterized in a solid-state as follow: intact, and PM and FD of FE/RMβCD complexes.

FT-IR was used to determine the interactions between guest (FE) and host (CD) molecules in the solid-state and to verify changes or shifts in the absorption spectra in CD complexation. The samples were measured by FT-IR spectrometer (Thermo Fisher model Nicolet iS10, Waltham, MA, USA) using the attenuated total reflectance (ATR) technique. The samples were analyzed at room temperature and the data were recorded in the range of 500–4000 cm^−1^.

PXRD was utilized for the determination of the crystalline structures of FE and CD, and for detection of CD complexation of FE in solid states. The PXRD patterns were recorded using a powder X-ray diffractometer (Rigaku^TM^ model MiniFlex II, Tokyo, Japan) and operated at a voltage of 30 kV and a current of 15 mA. The samples were analyzed at the 2θ angle range of 3° to 40° with the following process parameters: step size of 0.020° (2θ), and scan speed of 2° per minute.

DSC was used to study the thermal transitions involving heat capacity changes in the FE/RMβCD complexation. DSC thermograms were determined by a scanning calorimeter (Mettler Toledo^TM^, DSC822 STAR System, Giessen, Germany). The samples (3–5 mg) were heated (10 °C/min) in sealed aluminum pans under nitrogen. The temperature range was from 30 to 250 °C. An empty aluminum pan was used as a reference.

#### 3.5.2. Solution-State Characterization

^1^H-NMR spectroscopy was used to determine the difference in proton chemical shifts between the free guest (drug) or host (CD) and its complex (drug/CD). Pure solid samples of FE, RMβCD, and FE/RMβCD were dissolved in DMSO-*d_6_*. ^1^H-NMR spectroscopy measurements were performed using the BRUKER model AVANCE III HD, a 500 MHz ^1^H-NMR spectrometer (Bruker Corporation, MA, USA). The spectra and chemical shift values were recorded. DMSO-*d_6_* (2.500 ppm) served as an internal standard, and the chemical shift values were calculated according to the following formula:(4)$${\Delta \delta }^{*}=\left(\delta_{\mathrm{complex}}-\delta_{\mathrm{free}}\right)$$

Two-dimensional (2D) NMR spectroscopy (ROESY) was used to further investigate the binding mode of FE with RMβCD and to confirm the structure of the inclusion complexes characterized by ^1^H-NMR. This experiment was performed with a spectral width of 0–5500 Hz using 40 scans at 25 °C. The acquisition time was 0.1249 s; relaxation delay, 2.0 s; and spin-lock mixing time, 200 ms. 2D NMR spectra were recorded for the 1:1 M ratio of FE/RMβCD complex, and DMSO-*d_6_* was used as an internal reference.

### 3.6. Molecular Docking and Molecular Dynamic Simulation of the FE/RMβCD Complex 

The 3D structure of FE was downloaded from the PubChem database (compound ID: 3339). The structure of βCD was extracted from the crystal structure of the Protein Data Bank (PDB ID: 1G1Y). RMβCD was constructed by attaching methyl groups to βCD, and the 3D structure was constructed using the ChemDraw program. The FE and RMβCD structures were added to explicit hydrogen atoms, minimized using MarvinSketch version 22.3 (evaluation/non-commercial license), and converted to a PDB format using OpenBabel version 3.1.0 [74]. To predict the configuration of the FE/RMβCD complex, a docking environment was set using the BioTite Python package version 0.33.0 [75] with a search space of 20 Å × 20 Å × 20 Å around the center of the RMβCD molecule. Docking was performed by vina application [76] implemented in BioTite with default parameters. The resulting docking modes were ranked by energy score, clustered using root-mean-square deviation (RMSD), and visualized.

The docking results were used as the starting molecular configuration of the molecular dynamic simulation. The FE and RMβCD structures were separated from the docking results provided in PDB format, converted to mol2 format using MarvinSketch, and submitted to the SwissParam web service [77] to create the topology and parameters in the CHARMM36 force field version July 2021 [78]. The topology of FE and RMβCD were patched and combined to form a complex using PFSgen version 2.0, solvated using TIP3 water [79] 15 Å in each direction, and ionized using Visual Molecular Dynamic (VMD)’s solvate and autoionized plugins (version 1.9.2) [80]. The simulations were performed using Nanoscale Molecular Dynamics (NAMD) version 2.14 [81]. The complex was minimized for 1000 steps and simulated for 50,000 steps, with 2 fs of each step using particle mesh Ewald and by Langevin dynamic simulation at a temperature of 300 K and pressure of 1 atm. The energy and RMSD of the molecular dynamic simulation were calculated using VMD’s NAMD Energy and RMSD Trajectory Tool plugins (version 1.9.2) [80].

### 3.7. Preparation of FE and FE/RMβCD-Loaded EuNPs

The FE-loaded EuNPs were prepared by the (O/W) solvent evaporation technique. FE and Eudragit^®^ RL 100 were briefly dissolved in 15 mL of acetone. The organic phase obtained was added dropwise to 30 mL of aqueous phase containing 0.1% (*w*/*v*) HPMC or 1% (*w*/*v*) PVA. In FE/RMβCD-loaded EuNPs, the water-in-oil-in-water (W_1_/O/W_2_) emulsion solvent evaporation method was employed with modifications [82]. First, 5 mL of internal aqueous phase (containing FE aqueous solution) in 5% (*w*/*v*) RMβCD was emulsified in 15 mL of acetone (containing dissolved FE in Eudragit^®^ RL 100), using a probe sonicator for 15 s. Then, the resulting primary emulsion was added dropwise to 25 mL of aqueous phase containing 0.1% (*w*/*v*) HPMC or 1% (*w*/*v*) PVA while being stirred. To generate the W_1_/O/W_2_ emulsion, it was sonicated again using a probe sonicator for 1 min under the same conditions. To remove the acetone, all preparations were continuously stirred at room temperature for 2–3 h. Finally, 0.1% (*w*/*v*) EDTA, 0.02% (*w*/*v*) BAC, and a sufficient amount of sodium chloride to achieve isotonicity were added to the obtained mixture.

To increase drug loading, FE/RMβCD-loaded EuNP eye drop suspensions were further developed. The preparation method was the same as described earlier, but the total amount of FE in the formulation was increased to 0.3–0.5% *w*/*v*. The resulting FE-loaded EuNP suspensions were heated in a sonicator at 60 °C for 30 min. Finally, the preparations were passed through a Microfluider LM 20, a high-pressure homogenizer (HPH) (Westwood, MA, USA), at 20,000 psi for 20 cycles. The HPH conditions were validated to determine the optimum particle size reduction. The composition of FE-loaded EuNP dispersions and suspensions are shown in Table 7.

### 3.8. Physicochemical and Chemical Characterizations of FE-Loaded EuNP Eye Drop Formulations

#### 3.8.1. Appearance, pH, Osmolarity, Viscosity

The appearance of FE-loaded EuNP eye drop formulations was subjected to visual inspection. The pH values of the formulations were measured using SevenCompact™, a pH meter (Mettler Toledo, Gießen, Germany), at 25 °C. Using the freezing point depression principle, osmolality was measured using OSMOMAT 3000 basic, an osmometer (Gonotec GmbH, Berlin, Germany), at 25 °C. The viscosity of the formulation was measured using Sine-wave Vibro, a viscometer (A&D Company, Limited, Tokyo, Japan). The tuning-fork vibration method was applied at a frequency of 30 Hz and temperatures of 25 °C and 34 °C. Each sample was analyzed in triplicate.

#### 3.8.2. Particle Size and Size Distribution, and Zeta Potential

The particle size, size distribution, and zeta potential values of the samples were measured by the DLS technique (Zetasizer^TM^ Nano ZS software version 7.11; Malvern, UK). The FE-loaded EuNP dispersions were placed in a cuvette and in the instrument. The measurements were performed at a scattering angle of 180° and a temperature of 25 °C. In the FE-loaded EuNP suspensions, prior to determination, the samples were centrifuged at 3500 rpm for 40 min to separate these into two portions, i.e., the supernatant and solid particle fractions. The supernatant fraction was withdrawn and analyzed. All measurements were performed in triplicate. The particle size and size distribution of the solid fraction were determined by optical microscopy (Eclipse E200; Nikon^TM^, Tokyo, Japan) according to USP General Chapter <776> [83].

#### 3.8.3. Total FE Content and EE Analysis

The total FE content of the eye drop formulations was determined by diluting 100 µL of the sample with a mixture of acetonitrile and water (70:30, *v*/*v*) and was quantified by HPLC. To determine the EE, the formulations were ultracentrifuged at 18,000 rpm and 4 °C for 1 h. The dissolved FE in the supernatant was quantified by HPLC after appropriate dilution with the same solvent. All measurements were performed in triplicate. The %EE was calculated as follows:(5)$$\%\mathrm{EE}=\frac{\left[D_{t}-D_{s}\right]}{\left[D_{t}\right]}\times 100$$
where D_s_ is the FE content in the supernatant and D_t_ is the total FE content.

### 3.9. In Vitro Mucoadhesive Study

The mucoadhesive characteristics of FE-loaded EuNP formulations were determined by a modified method [84]. First, 0.1% *w*/*v* aqueous mucin solution (from porcine stomach Type II) was briefly prepared in a simulated tear fluid at a pH of 7.4 (composition [100 mL]: 0.68 g NaCl, 0.22 g NaHCO_3_, 0.008 g CaCl_2_·2H_2_O, and 0.14 g KCl. The FE eye drop formulations were mixed with 1 mL of mucin solution, incubated at 35 °C for 30 min, and kept at room temperature for 24 h. The samples were ultracentrifuged at 18,000 rpm and 4 °C for 1 h. The supernatant was collected, and free mucin was quantified by a UV-1800 spectrophotometer (Shimadzu, Tokyo, Japan) at 251 nm. All ingredients contained in the formulation were also scanned at 251 nm and it was found that there was no interference at the mucin UV wavelength. The binding efficiency (%mucoadhesion) of mucin with EuNPs was calculated using the following equation:(6)$$\%\;\mathrm{Mucoadhesion}=\frac{\mathrm{Total}\;\mathrm{mucin}\;\mathrm{concentration}-\mathrm{Mucin}\;\mathrm{concentration}\;\mathrm{in}\;\mathrm{the}\;\mathrm{supernatant}}{\mathrm{Total}\;\mathrm{mucin}\;\mathrm{concentration}}\times 100$$

### 3.10. In Vitro Release Studies

The in vitro permeation of FE/RMβCD-loaded Eudragit^®^ NP formulations through a semipermeable membrane (MWCO 12–14 kDa) was determined by a modified Franz diffusion cell. Phosphate buffer saline, pH 7.4 with 1% (*w*/*v*) RMβCD was used as the receptor medium. CD was added to the receptor phase (12 mL) to maintain a sink condition. A sample (1.5 mL) of each formulation was placed on the donor phase. The study was conducted at 35 ± 0.2 °C and the receptor phase was stirred continuously at 150 rpm during the experiment. A 150 µL aliquot of the receptor medium was withdrawn at various time intervals (1, 2, 3, 4, 5, 6, 8, and 12 h) and replaced immediately with an equal volume of fresh receptor medium. The amount of FE in the receptor medium was determined by HPLC and the amount of cumulative drug release was calculated. Each formulation was conducted at least in triplicate. To investigate the drug release kinetics, in vitro release data were fitted into a zero order, first order, Higuchi, and Korsmeyer–Peppas model.

### 3.11. In Vitro Permeation Studies

#### 3.11.1. Preparation of Octanol Dual Membrane and Mucin-Coated Octanol Membrane

An octanol dual membrane (i.e., artificial membrane) was prepared according to the method reported by Soe et al. (2020) [85]. First, a 4% (*w*/*v*) CA solution was briefly prepared by dissolving CA in acetone. CA was used to create the supporting matrix of the octanol membrane. The resultant CA solution was diluted with 6 mL of ether and ethanol solution (8.5:1.5 *v*/*v*), added to 4 mL of *n*-dodecanol, and mixed thoroughly to prepare the coating solution. The cellophane membrane with 12–14 kDa MWCO was cut into 4 cm × 4 cm squares and hung vertically on a string. The fused membrane was made by pouring 1.5 mL of the coating solution over the cellophane membrane at 15-min intervals. Coating was repeated three times, and the dual membrane was allowed to dry overnight. On the other hand, 1.5 mL of aqueous mucin solution (0.1% *w*/*v*) was layered onto the prepared octanol dual membrane to create the mucin-coated membrane. The membranes were dried for 2–3 h.

#### 3.11.2. In Vitro Permeation of FE-Loaded EuNP Eye Drop Formulations through Artificial Membranes

A modified Franz diffusion cell apparatus was used to determine the in vitro permeation of FE/RMβCD-loaded EuNP formulations through artificial membranes (uncoated and coated with mucin). Phosphate-buffered saline (PBS) (pH 7.4) with 1% *w*/*v* RMβCD was used as the receptor medium. RMβCD was added to the receptor phase to provide a sink condition. A 1.5 mL sample of each formulation was placed in the donor phase. The study was conducted at 35 ± 0.2 °C, and the receptor phase was continuously stirred at 150 rpm. Then, a 150 µL aliquot from the receptor medium was withdrawn and immediately replaced with an equal volume of fresh receptor medium at various time intervals (1, 2, 3, 4, 5, 6, 8, and 12 h). The amount of FE in the receptor medium was determined by HPLC. Each formulation was tested in triplicate. Flux (*J*) was calculated from the linear part of each permeability profile, and the apparent permeation coefficient (*P*_app_) was determined using Equation (7).
(7)$$J=\frac{dq}{A\cdot dt}=P_{\mathrm{app}}\cdot C_{d}$$
where *A* is the surface area of the mounted membrane 1.7 cm^2^ and *C_d_* is the initial concentration of the drug in the donor chamber. Steady-state flux was calculated as the slope of the linear plot on the amount of drug in the receptor chamber (*q*) versus time (*t*).

### 3.12. Morphological Characterizations

#### 3.12.1. TEM Analysis

To observe the morphology and confirm the particle size values of the formulation obtained by DLS, selected FE-loaded EuNP suspensions (F6 and F7) were analyzed by TEM. Initially, the FE eye drop formulations were centrifuged at 3500 rpm for 40 min to separate the supernatant and solid particle portions. The supernatant obtained from the samples was placed on a Formvar-coated grid, which was blotted with filter paper and transferred onto a drop of negative stain. An aqueous solution of 1% uranyl acetate was used for negative staining. The samples were air-dried at room temperature and examined by TEM (Model JEM-1400, JEOL, Peabody, MA, USA).

#### 3.12.2. SEM Analysis

First, the solid parts of the samples obtained from Section 3.12.1 were placed on a slide and dried overnight in a desiccator at room temperature. Subsequently, these were mounted on stubs and coated with a thin layer of gold under argon atmosphere at room temperature. Finally, the surface morphology of the solid particles in the FE-loaded EuNP suspensions was observed by SEM (JSM-7610F; JEOL, Peabody, MA, USA).

### 3.13. In Vitro Hemolytic Study

Sheep blood was collected and centrifuged at 3000 rpm for 10 min, and the supernatant was pipetted off. To obtain the initial volume, the red blood cells (RBCs) were washed three times and resuspended in PBS with a pH of 7.4. The RBCs were counted using a hemocytometer (Boeco, Hamburg, Germany) after proper dilution. To compare the effects of encapsulating FE NP, FE/RMβCD complexes (saturated FE in 5% *w*/*v* RMβCD) were included in the study. The samples were added to the resuspended RBCs, and the resulting suspensions were diluted with PBS to obtain final concentrations (ranging from 10 to 1000 μg/mL FE equivalents). The drug-free EuNP formulations were also mixed with RBCs and diluted with PBS in the same concentration range as their respective FE formulations. The samples were placed in an NB-205, a shaking incubator (N-Biotek, Gyeongggi-do, Korea), at 100 rpm and 37 °C for 30 min, and placed in an ice bath to stop hemolysis. Unlysed RBCs were removed by centrifugation (Thermo Fisher Scientific, Model X3, Waltham, MA, USA) at 3000 rpm for 5 min. The supernatant was transferred to a 96-well plate, and hemoglobin was measured at 576 nm using a microplate reader (CALIO star) [86]. The percentage of hemolyzed RBCs was determined using the following equation:(8)$$\%\mathrm{Hemolysis}=100\;\frac{\left(\mathrm{Abs}-{\mathrm{Abs}}_{0}\right)}{\left({\mathrm{Abs}}_{100}-{\mathrm{Abs}}_{0}\right)}$$
where Abs, Abs_0_, and Abs_100_ are the absorbance of the sample, PBS control, and distilled water, respectively.

### 3.14. HET-CAM

The HET-CAM assay was performed using fertile broiler chicken eggs, according to the ICCVAM-recommended test method protocol: HET-CAM test method [87]. First, eggs were incubated for nine days in an automatic rotation incubator (Machine 4 Biz, Bangkok, Thailand) at 38.0 ± 0.5 °C and 58.0 ± 2.0% relative humidity. During the last 24 h of incubation, the rotation was stopped to obtain the air sac located in the wider part of the egg. On the 9th day of incubation, the eggshells were opened (air chamber side) to remove the inner membrane without damaging the vasculature. Subsequently, 300 µL of the tested samples, as well as the negative control (C−) (i.e., 0.9% *w*/*v* sodium chloride solution) or positive control (C+) (i.e., 0.1% *w*/*v* sodium hydroxide solution), was directly applied to the chorioallantoic membrane (CAM) surface. The CAM was observed for 0.5, 2, and 5 min. Irritation scores were marked from 0 to 21 according to Luepke’s procedure (1985) [88] and irritation was classified as (I) hemorrhage (bleeding from the vessels), (II) vascular lysis (blood vessel disintegration), and (III) coagulation (intra- and/or extravascular protein denaturation). The experiment was performed in triplicate.

### 3.15. CV and STE Test

The MTT assay was used to investigate in vitro cytotoxicity [89,90]. This method can be used to assess the possibility of eye discomfort in a tested sample [91]. The selected FE/RMβCD-loaded EuNP suspension (F6) was briefly tested for toxicity in the SIRC cell line (CCL-60; ATCC, Manassas, VA, USA). A complete medium containing Eagle’s minimum essential medium and fetal bovine serum was used to dilute each sample to concentrations of 1–400 µg/mL. The cells were grown in complete media, kept at 37 °C in a 5% CO_2_ humidified air incubator, and seeded in 96-well plates at a density of 1 × 10^5^ cells/well/100 µL for 24 h. Each 100 µL test sample was added to a well. The cells were cultured for 24 h before being rinsed twice with PBS (pH 7.4). Each well was filled with an MTT solution in PBS (pH 7.4) and incubated for 4 h. Formazan crystals were dissolved in isopropanol (100 µL/well) with 0.04 M HCl. FLUOstar Omega, a microplate reader (BMG Labtech, Ortenberg, Germany), was used to measure the optical density (OD) of each well at 570 nm. CV was computed using Equation (9). If the %CV was <70%, then the test samples were classified as lethal to cells.
(9)$$\%\mathrm{CV}=100\;\times \;\frac{{\mathrm{OD}}_{\mathrm{sample}}}{{\mathrm{OD}}_{\mathrm{control}}}$$
where OD_sample_ and OD_control_ are the ODs of the media from wells containing SIRC cells treated with the samples and MTT solution and those incubated with only MTT solution, respectively.

The in vitro ocular irritation test was performed in accordance with Takahashi’s STE test technique. After a 5-min exposure to 200 µL of either a 5% or 0.05% test sample dispersed in normal saline, the SIRC cells were evaluated for %CV [71]. The potential of the STE test for ocular irritation was evaluated using the corresponding irritation scoring criteria. The results from using 5% and 0.05% test samples were summated to rank the eye irritation potential. “Minimal ocular irritant”, “moderate ocular irritant”, and “severe ocular irritant” were defined as a total score of 1, 2, and 3, respectively.

### 3.16. Statistical Analysis

All quantitative data are presented as the mean ± standard deviation (SD). Means were evaluated for the statistical significance of differences by one-way ANOVA with Tukey’s post hoc test, using the SPSS software version 16.0. Statistical significance was set at *p* < 0.05.

## 4. Conclusions

FE is a poor water-soluble compound with a physiological pH. The solubility of FE was enhanced by inclusion complex formation with RMβCD. FE was deeply inserted into the RMβCD cavities at stoichiometric ratios of 1:1 and 1:2. The addition of water-soluble polymers (i.e., PVA and HPMC) to the FE/RMβCD complex provided an additive effect on CD solubilization. We successfully formulated PVA-stabilized FE/RMβCD-loaded EuNP eye drops. Specifically, they had good physicochemical properties, good mucoadhesion property, sustained-release properties, and low %hemolysis. Further, the eye drops were classified as a moderate eye irritant, causing only mild irritation, and were cytocompatible with SIRC cell lines. However, in vivo pharmacokinetic studies on rabbit eyes, particularly in the posterior segment of the eye, and cellular anti-VEGF need to be considered in future prospective studies.

## Figures and Tables

**Figure 1 molecules-27-04755-f001:**
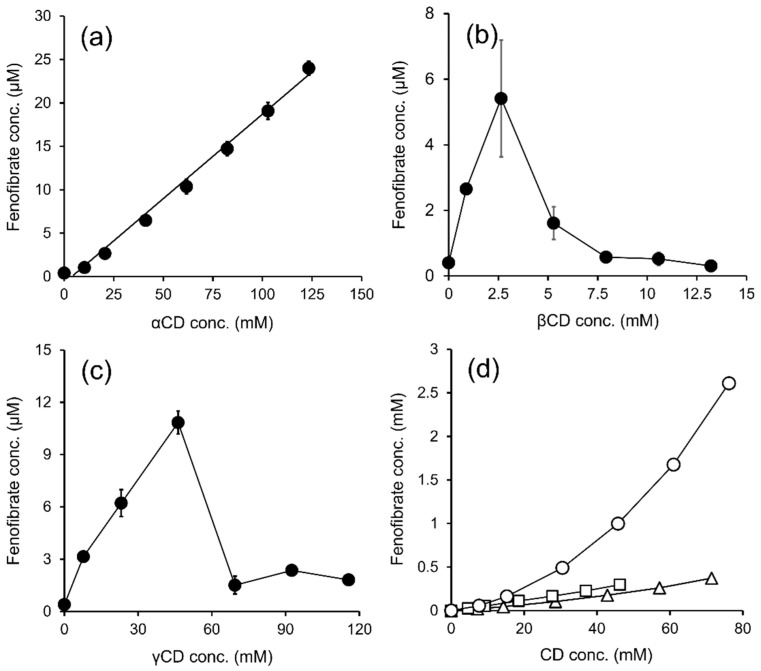
The phase-solubility profiles of fenofibrate in different aqueous CD solutions, namely αCD (**a**), βCD (**b**), γCD (**c**), and βCD derivatives (**d**), which include HPβCD (△), SBEβCD (◻), and RMβCD (◯).

**Figure 2 molecules-27-04755-f002:**
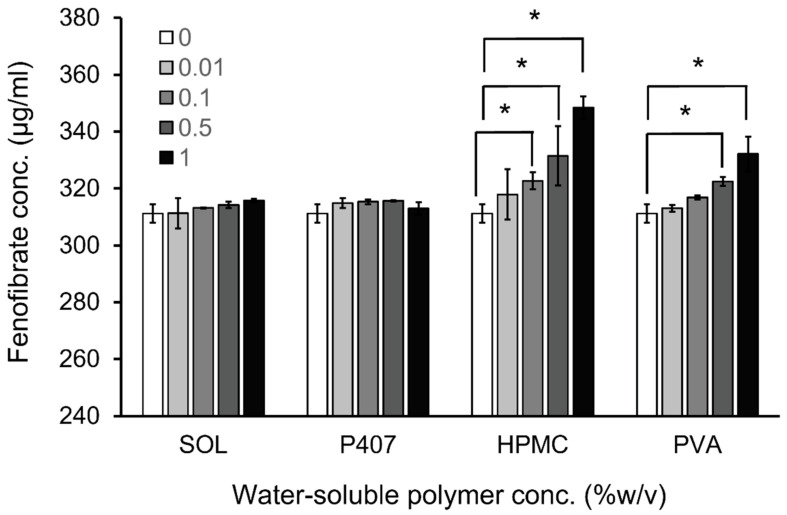
The solubility of fenofibrate in aqueous 5% *w*/*v* RMβCD solutions containing various concentrations of water-soluble polymers (0–1.0% *w*/*v*). SOL, Soluplus^®^; P407, poloxomer 407; HPMC, hydroxypropyl methylcellulose; PVA, polyvinyl alcohol. * Statistical difference (*p* < 0.05) when compared to that of solutions without polymers.

**Figure 3 molecules-27-04755-f003:**
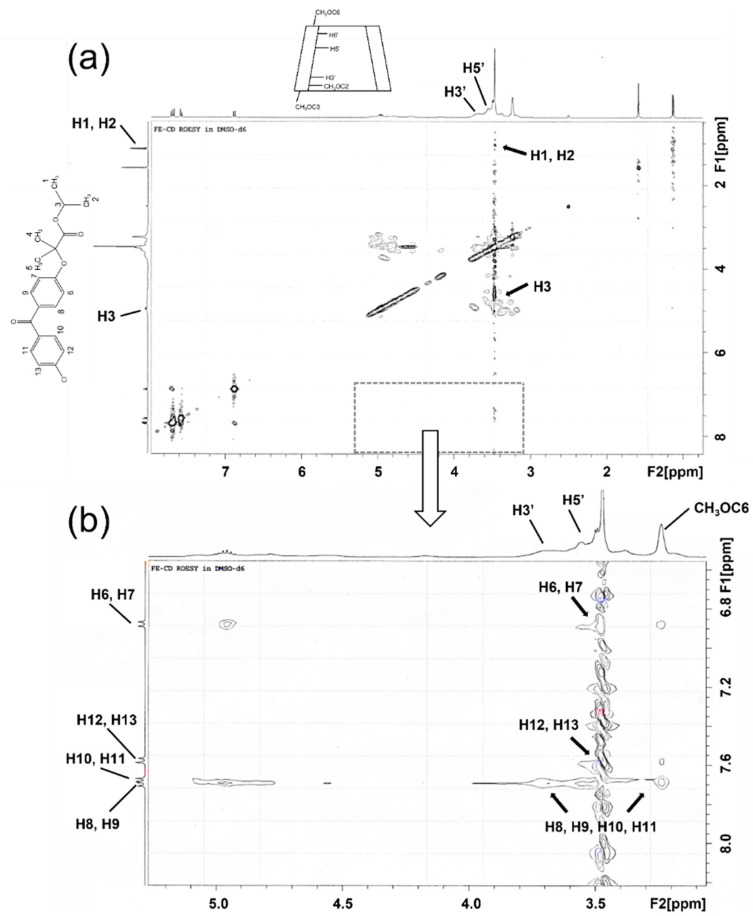
2D ROESY spectrum of the FE/RMβCD complex in DMSO-*d*_6_. The cross peaks of FE protons and inner protons of the RMβCD cavity are shown in two views: (**a**) the whole region and (**b**) the expanded region.

**Figure 4 molecules-27-04755-f004:**
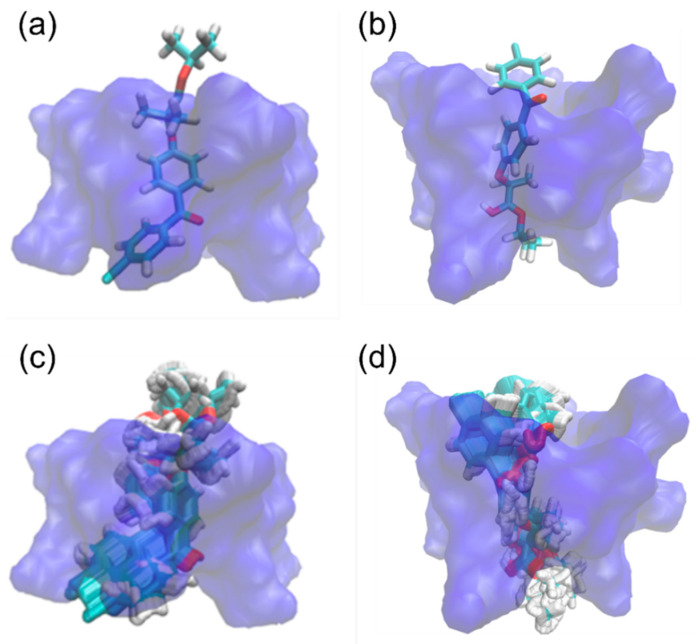
Based on molecular docking, the binding orientations of the FE/RMβCD complex were labelled as form I (**a**) and form II (**b**). The surface representation in blue denotes the RMβCD molecule, and the licorice representation denotes the FE molecule. Meanwhile, the movements of FE in the RMβCD cavity are labelled as form I (**c**) and form II (**d**). The surface representation in blue denotes the RMβCD molecule, and the licorice representation denotes the FE molecule.

**Figure 5 molecules-27-04755-f005:**
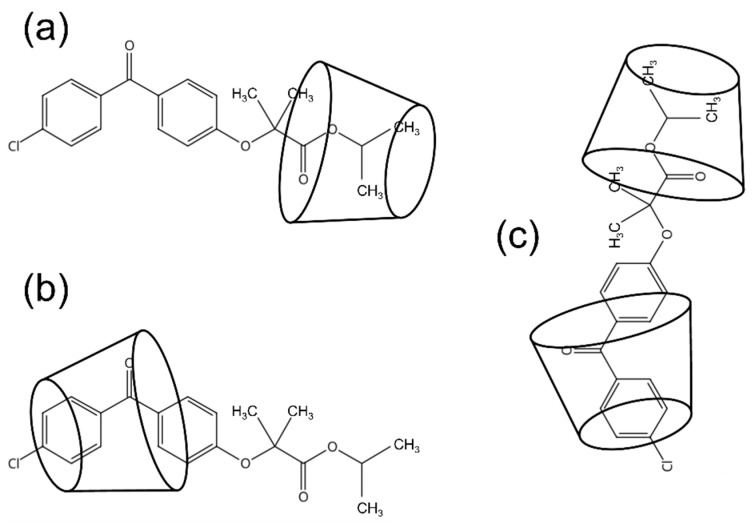
The proposed conformation of 1:1 FE/RMβCD complex (**a**) at the isopropyl group insertion of FE, (**b**) aromatic ring insertion of FE, and (**c**) 1:2 FE/RMβCD complex.

**Figure 6 molecules-27-04755-f006:**
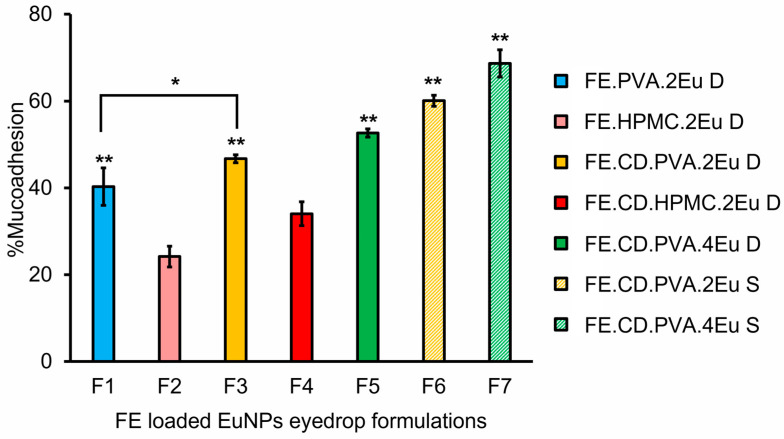
Percentage of mucoadhesion in FE-loaded EuNP eye drop formulations. D, dispersion; S, suspension. * Statistical difference (* *p* < 0.05, ** *p* < 0.01) when compared to the corresponding HPMC-based formulations (F2 or F4).

**Figure 7 molecules-27-04755-f007:**
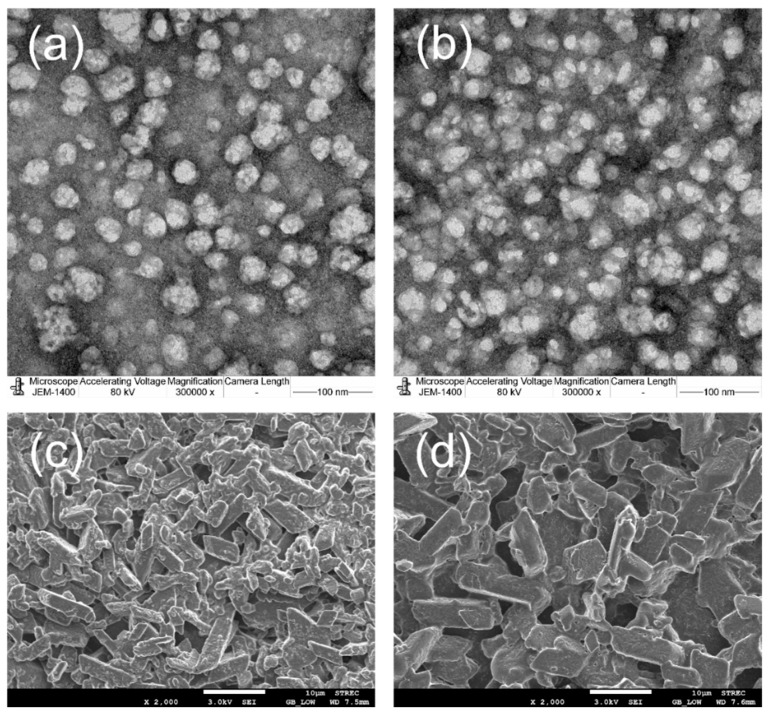
The TEM photographs of FE/RMβCD-loaded EuNPs, particularly (**a**) F6 and (**b**) F7, at magnification (100 nm bar). The SEM photographs of FE/RMβCD-loaded EuNPs, particularly (**c**) F6 and (**d**) F7, at magnification (×2000).

**Figure 8 molecules-27-04755-f008:**
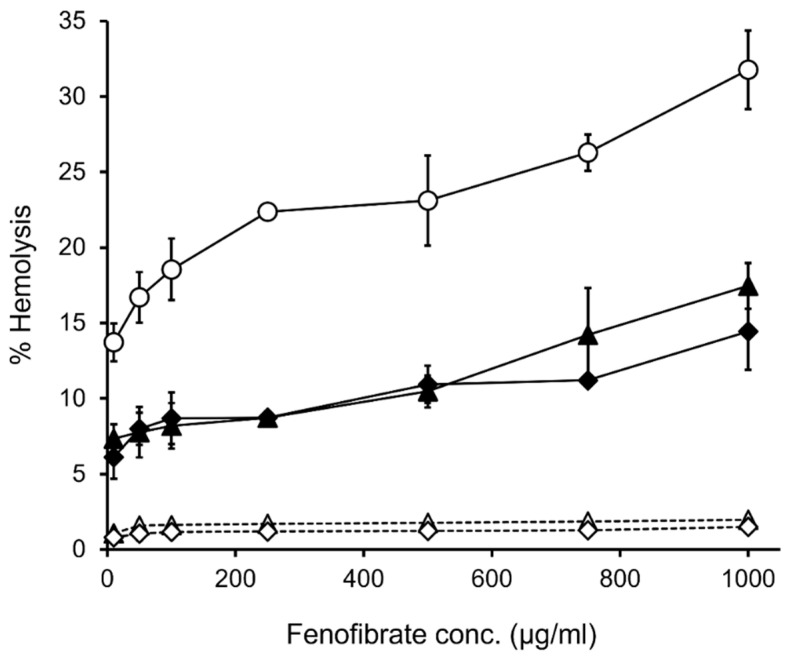
In vitro hemolytic study of sheep RBC at various concentrations of FE. FE/RMβCD complexes (◯), F6 (◆), F7 (▲), drug-free F6 (◇), drug-free F7 (△). *n* = 3. The data are presented as mean ± SD.

**Figure 9 molecules-27-04755-f009:**
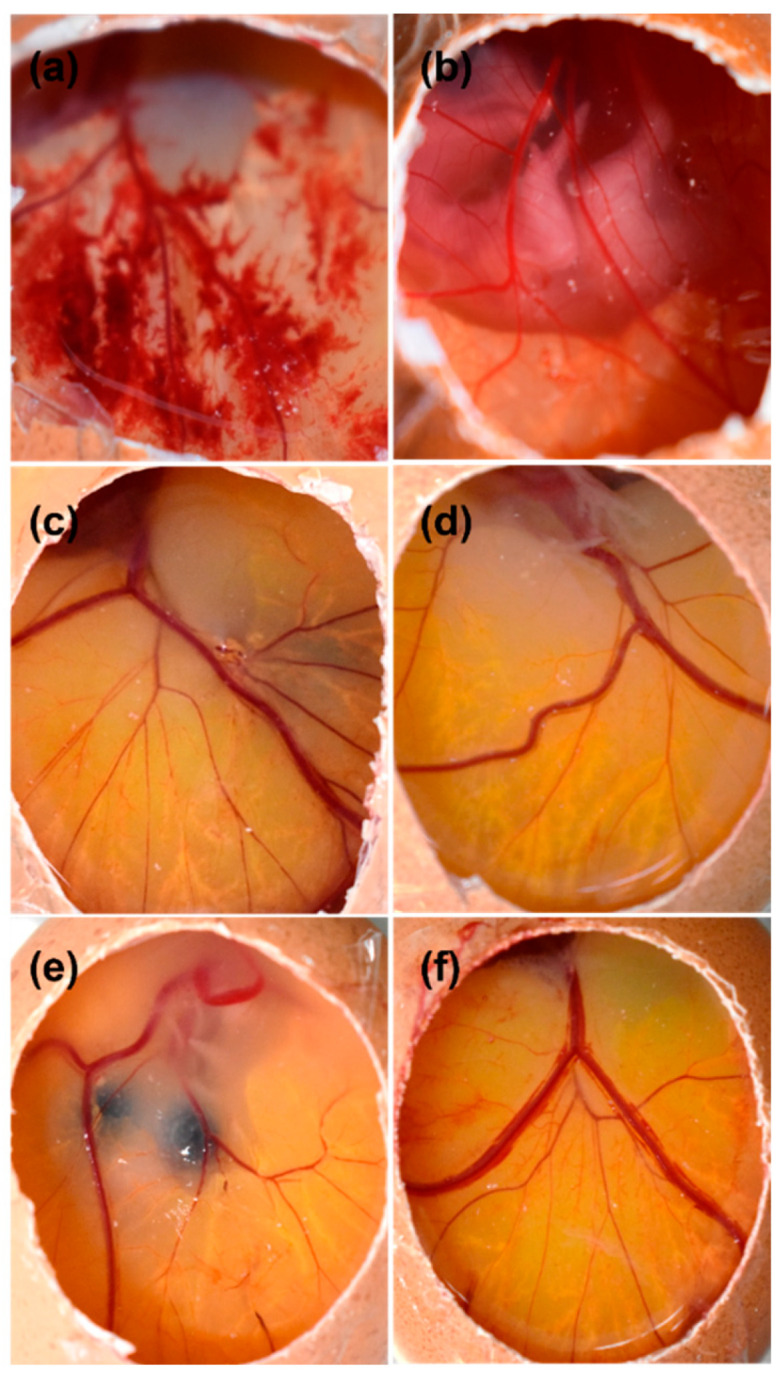
Images of the hen’s egg test on chorioallantoic membranes (HET-CAM) 5 min post-instillation of different samples, namely (**a**) NaOH (C+), (**b**) NaCl (C−), (**c**) drug-free F6, (**d**) drug-free F7, (**e**) F6, and (**f**) F7.

**Figure 10 molecules-27-04755-f010:**
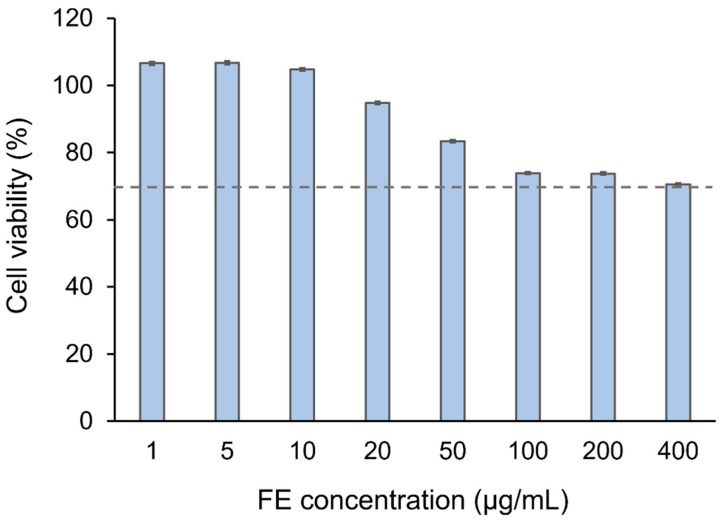
CV of SIRC cells after incubation with FE eye drop suspensions (F6).

**Table 1 molecules-27-04755-t001:** The percentage of FE remaining in aqueous 2.5% *w*/*v* and 5% *w*/*v* HPβCD solutions after zero to three heating cycles in an autoclave or ultrasonic bath.

No. of Cycles	Percentage of FE Content	
	HPβCD (2.5% *w*/*v*)	HPβCD (5% *w*/*v*)
*Autoclaving* ^a^		
cycle 1	92.29 ± 1.96	93.99 ± 1.15
cycle 2	91.24 ± 2.40	92.76 ± 0.94
cycle 3	89.77 ± 1.41	92.35 ± 0.90
*Sonication* ^b^		
cycle 1	99.91 ± 1.92	102.06 ± 0.65
cycle 2	99.09 ± 0.52	100.54 ± 1.75
cycle 3	99.32 ± 0.49	101.27 ± 0.96

^a^ Each cycle was performed at 121 °C for 20 min. ^b^ Each cycle was performed at 60 °C for 30 min. *n* = 3. The data are presented as mean ± SD.

**Table 2 molecules-27-04755-t002:** The apparent stability constants (K_1:1_, K_1:2_) and CE of FE/CD complexes in aqueous solutions at 30 ± 1 °C.

Cyclodextrin	Type	R^2^	K_1:1_ (M^−1^)	K_1:2_ (M^−1^)	CE (×10^−3^)
αCD	A_L_	0.993	500.7	-	0.20
βCD	B_s_	0.983	4638.6	-	1.85
γCD	B_s_	0.989	546.6	-	0.21
SBEβCD	A_L_	0.998	1598.3	-	0.63
HPβCD	A_P_	0.997	6571.9	13.43	2.62
RMβCD	A_P_	0.999	11,380.7	84.33	4.54

**Table 3 molecules-27-04755-t003:** The ^1^H-chemical shifts of FE alone and in the presence of RMβCD.

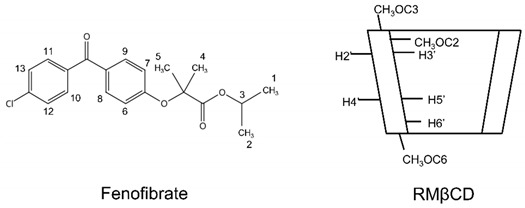			
^1^H	δ_(free)_	δ_(complex)_	Δδ* = δ_(complex)_ − δ_(free)_
*FE*			
H1, H2	1.141	1.139	−0.002
H3	4.967	4.965	−0.002
H4, H5	1.592	1.591	−0.001
H6, H7	6.887	6.888	+0.001
H8, H9	7.719	7.718	−0.001
H10, H11	7.693	7.693	0.000
H12, H13	7.605	7.608	+0.003
*RM* *β* *CD*			
H1′	4.986	- ^a^	- ^a^
H3′	3.701	3.700	−0.001
H5′	3.535	3.537	+0.002
H6′	3.488	3.486	−0.002
CH_3_OC6	3.240	3.238	−0.002

^a^ Could not be determined due to overlap with other signals

**Table 4 molecules-27-04755-t004:** Physicochemical and chemical properties of the FE-loaded EuNP eye drop formulations.

Formulations	pH	Osmolality	Viscosity	Z-Average	Size Distribution	Zeta Potential	%EE
		(mOsm/kg)	(mPa·s)	(d·nm)	(PDI)	(mV)	
F1	7.43 ± 0.01	312 ± 5	2.53 ± 0.02	75.29 ± 12.16	0.23 ± 0.03	+26.93 ± 1.50	48.05 ± 3.97
F2	7.44 ± 0.03	318 ± 3	2.59 ± 0.01	98.81 ± 13.43	0.28 ± 0.02	+26.32 ± 0.94	44.23 ± 0.39
F3	7.44 ± 0.04	315 ± 6	2.62 ± 0.02	53.35 ± 1.03	0.22 ± 0.03	+25.77 ± 1.66	77.74 ± 0.47
F4	7.48 ± 0.01	321 ± 2	2.73 ± 0.01	57.91 ± 5.24	0.22 ± 0.01	+22.23 ± 2.67	68.52 ± 0.81
F5	7.41 ± 0.02	318 ± 5	3.29 ± 0.08	60.65 ± 0.30	0.22 ± 0.01	+33.22 ± 2.06	83.59 ± 0.44
F6	7.43 ± 0.02	317 ± 1	3.09 ± 0.03	56.23 ± 0.84	0.21 ± 0.05	+28.11 ± 0.57	81.84 ± 1.34
F7	7.47 ± 0.01	319 ± 1	3.44 ± 0.03	62.62 ± 2.91	0.21 ± 0.02	+41.92 ± 1.10	87.04 ± 1.81

*n* = 3. The data are presented as mean ± SD.

**Table 5 molecules-27-04755-t005:** The *J* and *P*_app_ of FE-loaded EuNP eye drop formulations through artificial membranes.

Formulations	Octanol Dual Membrane		Mucin-Coated Octanol Membrane	
	*J* (µg·h^−1^·cm^−2^)	*P*_app_ (×10^−6^ cm·s^−1^)	*J* (µg·h^−1^·cm^−2^)	*P*_app_ (×10^−6^ cm·s^−1^)
F6	1.12 ± 0.27	0.31 ± 0.06	1.02 ± 0.20	0.28 ± 0.02
F7	2.32 ± 0.06 *	0.35 ± 0.02	0.97 ± 0.15 **	0.14 ± 0.02 *^,^**

* Statistical difference when compared to that of F6 (*p* < 0.05); ** statistical difference when compared to that of octanol dual membrane (*p* < 0.05). *n* = 3. The data are presented as mean ± SD.

**Table 6 molecules-27-04755-t006:** Scores obtained from the STE test of F6.

Concentration of the Formulation	% CV of SIRC Cells ^a^	Criteria for Scoring	Obtained Scores
5%	66.31 ± 0.02	If CV > 70%, then score = 0	1
		If CV ≤ 70%, then score = 1	
0.05%	80.04 ± 0.05	If CV > 70%, then score = 1	1
		If CV ≤ 70%, then score = 2	
		Total scores	2

^a^*n* = 4, mean ± SD.

**Table 7 molecules-27-04755-t007:** The composition of FE-loaded EuNP dispersions and suspensions.

Ingredients (% *w*/*v*)	Formulation ^a^						
	*Dispersions*					*Suspensions*	
	F1	F2	F3	F4	F5	F6	F7
FE	0.05	0.05	0.1	0.1	0.15	0.3	0.5
RMβCD	-	-	5	5	5	5	5
Eudragit^®^ RL 100	2	2	2	2	4	2	4
PVA	1	-	1	-	1	1	1
HPMC	-	0.1	-	0.1	-	-	-

^a^ Each formulation was composed of 0.1% EDTA and 0.02% BAC as an antioxidant and preservative, respectively. The pH was adjusted to 7.4 with 1N sodium hydroxide, and with sodium chloride to obtain isotonicity.

## Data Availability

Not applicable.

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
