# Peer review of "Development of Fenofibrate/Randomly Methylated β-Cyclodextrin-Loaded Eudragit® RL 100 Nanoparticles for Ocular Delivery"

_molecules, 2022, doi:10.3390/molecules27154755_

Round 1
Reviewer 1 Report
The paper "Development of Fenofibrate/Randomly Methylated ꞵ-Cyclodextrin-Loaded Eudragit® RL 100 Nanoparticles for Ocular Delivery" correlates well with the objectives and scope of the journal Molecules. The authors developed a formulation for ocular delivery based on Eudragit nanoparticles carrying inclusion complexes with cyclodextrins. It is an exciting work for researchers dedicated to drug delivery, supramolecular chemistry, and specialists in the approach to ocular pathologies. However, to improve the scope and understanding of the publication, the following major modifications should be made:
-Although the authors reported a large number of experimental results, in the opinion of this reviewer, additional assays should be performed in order to finish corroborating the hypothesis put forward:
1) X-Ray powder diffraction of the drug, excipients, inclusion complexes, and nanoparticles to study the crystalline state of the drug, corroborate that inclusion complexes are still formed within the nanoparticle, and understand the degree of non-included drug.
2) FTIR to verify possible intermolecular interactions between the components of the nanoformulation.
3) In the conclusion, the authors mention that there is a controlled release of the active ingredient, although they do not verify this experimentally. They must carry out Drug Release studies in biorelevant media to verify that the drug is indeed released and to analyze their differences with respect to inclusion complexes. On the other hand, the authors should mention the differences between RL and RS Eudragits in permeability and controlled release.
4) In the conclusion, the authors mention that the formulation has a prolonged ocular residence time. However, they did not verify this in the eye but only evaluated mucoadhesion in vitro. Correct this point
5) Figure 9 e shows a stain that could be due to vascular damage due to formulation F6. Since this is the formulation selected for subsequent trials, the authors should show another image or clarify what the stain is due to.
6) It would be necessary to add a table or scheme that clearly indicates which are the F1-F7 formulations since it becomes difficult for readers to understand the differences during the reading of the text.
-References missing:
1) Examples of Eudragit nanoparticles that have improved the biopharmaceutical properties of drugs should be included in the introduction. Suggested reference: https://doi.org/10.1016/j.ijpharm.2022.121594
· 2) The advantages of combining cyclodextrins with nanomaterials should be highlighted. Suggested reference: DOI: 10.3390/pharmaceutics13122131
3) Line 82: references that have studied the non-toxicity of Eudragit should be included.
4) Line 107: Include reference on thermal degradation of the drug.
Reviewer 2 Report
The article entitled “Development of Fenofibrate/Randomly Methylated ꞵ-Cyclodextrin–Loaded Eudragit® RL 100 Nanoparticles for OcularDelivery" shows a complex solubility improvement strategy and physicochemical and in vitro test investigation of fenofibrate as a BCS II drug. Although the test results presented show a complex and well-structured work, I believe that they need to be supplemented and reconsidered in many respects. My comments on this are summarized below.
- Perhaps one of the most critical points in the manuscript is that it is not stated what dose they intend to target with the fenofibrate formula. This is a particularly critical point because it would allow the solubility isotherms shown in Fig1 to be evaluated. The HPBCD, SBEBCD, and RAMEB derivatives shown in Fig1d offer the most efficient complexes. However, from an ophthalmological point of view, HPBCD is the most favorable because it has the most clinical experience with this CD derivative and RAMEB has a toxic effect at relatively low concentrations. On the other hand, enhanced complex stability (RAMEB), while increasing solubility to a greater extent, may at the same time reduce the permeability of the drug. In fact, the above can only be assessed on the basis of the target concentration / formulation concentration of the active substance (fenofibrate). Addressing the above in the corrected manuscript would certainly be important, and why RAMEB is more favorable than HPBCD.
- In Fig1, for easier traceability, it would be worthwhile to show 2.5 and 5% CD concentrations at the molar concentration in order to evaluate the fenofibrate concentration at subsequent RAMEB concentrations.
- The goal of thermal stability presented in Section 2.1 is only partially understood. In the case of a formulation, although stability during the preparation of the CD complex may be important, storage stability, possibly related to its use in vitro, would also be relevant. This is of particular interest in the use of CD, where one of the important effects may be a change in the stability conditions of the drug.
- The legend of Fig6 is undefined, difficult to understand and only based on a thorough review of the text. In my opinion, the figures should be traceable in themselves, so I suggest supplementing the explanation of the signs in the title of the figure.
- Another critical point in the paper is the in vitro permeability test presented in 2.6. First, it is not clear on what basis the presented model membrane can be considered suitable for modeling ophthalmic absorption. The choice of mucin is justified in my opinion, but the choice of octanol as a solvent membrane is, in my opinion, clearly flawed or unfounded. Although octanol as a partition solvent is excellent for determining lipophilicity (logP), it is particularly problematic in permeability studies. The main reason for this is that octanol causes increased membrane retention for most lipophilic compounds, which specifically inhibits the effective penetration of the compounds. An indication of this can also be seen here, as the Papp values given in Table5 are markedly lower. Normal permeability values range from 5 to 40, with values below 1 indicating particularly low permeability. Based on this, the significant nature of the Papp values obtained for the mucin-coated octanol membrane shown for F7 is statistically significant, but these differences do not constitute a relevant difference. In this Papp range, all values are considered low. However, it would be important to give a retention of fenofibrate membrane based on mass balance, which is expected to be very high. If this is the case then the values obtained are not relevant. Notwithstanding the above, it would still be important to explain why this permeability system is suitable for characterizing ophthalmic absorption.
- In 3.10.2 it would be important to give not only the flux but also the Papp equation / definition. Although it can be concluded from the description that it is given from the slope obtained from the change in concentration on the receiving side, it would be good to clarify this. Unfortunately, this method does not reveal the increased membrane retention that I described in the previous section of my review. Therefore, in a first approximation, it might be more advantageous to evaluate the formulation on the basis of effective permeability.
Reviewer 3 Report
Reviewer comment:
The manuscript entitled " Development of Fenofibrate/Randomly Methylated -Cyclodextrin–Loaded Eudragit® RL 100 Nanoparticles for Ocular Delivery " by Khin et al. is extremely interesting and the findings bring new information on the nanoparticles development.
The authors produced the drug-loaded nanoparticles using Eudragit a copolymer widely used in ophthalmic preparations plus cyclodextrin and water-soluble polymers. Undoubtedly, the nano- particulate preparations belong to the state-of-the-art technologies. A comprehensive characterization of the formulations has been made and described carefully and in detail.
Importantly, the permeability of the formulations is also addressed with octanol dual membrane and mucin-coated octanol membrane.
The article is well written, clear, fits the Pharmaceutics journal scope. So the reviewer's recommendation is in favor of a rapid publication, after addressing the following recommendations:
1. Introduction. All the polymers used in the study should be mentioned in the Introduction (possibly a short description and using simultaneously with CDs).
2. The structures of the studied substances with full and abbreviated names should be provided in the beginning of the manuscript for better visualization of the results (In the main document or possibly in SI file).
3. (Section 3.2, line 475; Section 3.3, lines 482-484) What were the reasons of the temperature 121 C in autoclave heating if the drug suspensions were heated in a sonicator at 60 °C for 30 min and allowed to cool at room temperature before the equilibration at 30 C? The reasons should be introduced.
4. (Section 3.2, line 484-486) Immediate filtration after reaching the equilibrium can result in a supersaturation solutions and solubility overestimation. Usually, the saturated solutions are subjected to filtration of centrifugation only after sedimentation (see Avdeef et al. ADMET & DMPK 4(2) (2016) 117-178; doi: 10.5599/admet.4.2.292). A detailed description of the procedures after reaching the equilibrium with the appropriate reference is required.
5. (Section 3.10.1, line 484-486). Octanol dual membrane and mucin-coated octanol membrane were prepared to evaluate the permeability of the formulations. How the membranes were checked for uniformity when prepared several times?
6. Since the water-soluble polymers capable of forming the micelles in solution (Soluplus, P407) an issue of CMC should be clarified concerning, firstly, the concentrations used and, secondly, the mutual influence of CD and polymer on the investigated properties of FE.

Round 2
Reviewer 1 Report
The Authors complies with all the suggestions made. The work is ready to be publish
Reviewer 2 Report
The manuscript is acceptable for publication in its present form.